# Estimation of the Filling Distribution and Height Levels Inside an Insulated Pressure Vessel by Guided Elastic Wave Attenuation Tomography

**DOI:** 10.3390/s21010179

**Published:** 2020-12-29

**Authors:** Robert Neubeck, Mareike Stephan, Tobias Gaul, Bianca Weihnacht, Lars Schubert, Arne Ulrik Bindingsbø, Jan-Magnus Østvik

**Affiliations:** 1Fraunhofer Institute for Ceramic Technologies and Systems (IKTS), Systems for Condition Monitoring, Maria-Reiche-Straße 2, 01109 Dresden, Germany; mareike.stephan@ikts.fraunhofer.de (M.S.); tobias.gaul@ikts.fraunhofer.de (T.G.); Bianca.Weihnacht@ikts.fraunhofer.de (B.W.); lars.schubert@ikts.fraunhofer.de (L.S.); 2Equinor ASA, Sandslivegen 90, NO-5254 Sandsli, Norway; abin@equinor.com; 3Department of Marine Technology, Norwegian University of Science and Technology (NTNU), Otto Nielsens Veg 10, NO-7491 Trondheim, Norway; 4Equinor ASA, Wessel Veg 41, NO-7501 Stjørdal, Norway; jostvi@equinor.com

**Keywords:** guided elastic waves, guided ultrasonic waves, high pressure separator vessel, filling distribution, height level detection, multi-interface measurement, attenuation tomography

## Abstract

The operation efficiency and safety of pressure vessels in the oil and gas industry profits from an accurate knowledge about the inner filling distribution. However, an accurate and reliable estimation of the multi-phase height levels in such objects is a challenging task, especially when considering the high demands in practicability, robustness in harsh environments and safety regulations. Most common systems rely on impractical instrumentation, lack the ability to measure solid phases or require additional safety precautions due to their working principle. In this work, another possibility to determine height levels by attenuation tomography with guided elastic waves is proposed. The method uses a complete instrumentation on the outer vessel shell and is based on the energy conversion rates along the travel path of the guided waves. Noisy data and multiple measurements from sparsely distributed sensor networks are translated into filling levels with accuracies in the centimeter range by solving a constrained optimization problem. It was possible to simultaneously determine sand, water, and oil phases on a mock-up scale experiment, even for artificially created sand slopes. The accuracy was validated by artificial benchmarking for a horizontal vessel, giving references for constructing an affordable prototype system.

## 1. Introduction

Separator vessels are generally objects with a cylindrical-shaped shell, vertically or horizontally aligned and consisting of normally high-grade materials. They divide various phases of oil and gas from the injected media by gravity. Depending on the architecture of the purifying process and the specific vessel type, the injected media can vary from a three-phase mixture of hydrocarbons, (salt–)water and solids directly extracted from the well/reservoir itself, till a pre-separated two-phase media mainly consisting of water, oil and gas of a cascaded process in later stages [1]. Since the recovered reservoir fluid itself is under high pressure, the first stages of the separation are as well, while with each processing stage the operation pressure is decreasing. Beside adjusting the operation pressure and temperature, the efficiency of a separation process can be controlled by the flow rate, adding of chemicals, drainage and finally by turn-around cycles. All those factors and therefore the complete operation efficiency and safety can profit from an accurate knowledge of the height levels in each separation phase.

Although in most operating countries the construction, the quality control and maintenance by Non-Destructive Evaluation (NDE) is regulated by the American Society of Mechanical Engineers (ASME) guidelines [2], the estimation of the filling level itself is not included. Nevertheless, various NDE methods can be found in the outlining literature [3,4]. Different techniques especially concerning separator vessels, summarizing their advantages and drawbacks are discussed in [3,5,6]. Based on those investigations, Table 1 illustrates an updated overview, including the following literature research.

In the context of separator vessels, displacers [4,7], pressure sensors [8,9], ultrasonic pulse-echo devices [4,7], electromagnetic radar [8], electric capacity methods [5,6,10,11,12] and various approaches of gamma ray methods [7,13,14,15,16] have proved their applicability for height evaluation.

Displacers are submerged floating bodies with a defined density between two phases of interest. If the transition level is changing, the resulting buoyant force is captured by a reading device. Multiple phase-levels can be detected, by adding multiple displacers with different densities. However, these devices tend to fail in the unexpected presence of foam or emulsion since their presence effects the pressure gradient along an interface [13]. Additionally, solid phases like sand, clay or wax cannot be monitored accordingly to the working principle. The last limitation applies for pressure sensors as well. Both methods are contact-based and need direct access to the vessels content, which is generally realized by additional bridles or bypasses, which could fail in recreating the exact conditions inside the vessel.

Other methods such as radar or ultrasonic pulse-echo devices need to be instrumented at least partly inside a vessel. Ultrasonic devices are generally based on time-of-flight measurements and are commonly used to measure the gas-liquid transition height [4,7]. However, the attenuation rates due to geometrical spreading and scattering losses of bulk waves are decreasing the penetration depth significantly and require an enhanced power source, if deeper phase transitions should be monitored. Moreover, their main disadvantage is that the scattering effect of rough surfaces as foam or emulsion are disturbing the measurements or at least attenuates the pulses significantly. On the other hand, lAlshaafi et al. [17] used this effect as a separate measurement contrast to track those emulsion layers. In contrast, radar applications are using electromagnetic waves in the gigahertz spectrum and are based on the contrast in the relative permittivity, which is relatively stable over a wide range of process temperature and pressure. Such a system is usually arranged as non-contact pulse radar [4], comparable to the ultrasonic measurements, or with an additional antenna in contact with the filling as guided radar [4,8]. The latter has proven its capability even for the solid phase, if the antenna is not exceeding a certain length and is free of residual products as wax or hardened clay.

Hjertaker et al. [7] provided the combination of electrical, ultrasonic, thermal and gamma ray methods for three-phase hydrocarbon separators and introduced the term of tomometry, meaning multipoint cross-sectional metering aiming to acquire information on the cross-sectional flow-component distribution. Skeie et Halstensen [8] chose a comparable approach by combining measurements of multiple fixed relative pressure sensors and a guided radar system using multivariate calibration to increase the accuracy. Arvoh et al. [9] proceeded the work with a single pressure sensor measuring while traversing from bottom to top of the vessel. Finally, the results of various regression schemes were compared with each other.

Despite the achieved success with each single measurement method, the combination of methods and elaborate processing schemes, there is so for no NDE standard that meet all requirements of the industry [3]. Especially instrumenting existing three-phase separator vessels without any additional change in their design such as bridles, holes or instrumentations inside remains a task which is currently only realizable by gamma radiation devices or Electric Capacity Tomography (ECT). Isaksen et Nordvedt [10,11] proposed various tomographic reconstruction techniques to estimate flow distributions in pipelines. Bukhari et Yang [5,6,12] added a calibration scheme, including low and high permittivity contrasts. Together with an a priori database the system can monitor air-oil and oil-water phase transitions in horizontal separators effectively. On the other hand, a static database implies that the filling content should not vary in ongoing measurements regarding its electrical properties, which can be critical in real operations. Also, ECT is not applicable to vertically aligned vessels without either loosing resolution or increasing the number of sensors and therefore the costs. Especially the latter impeded the technology from the commercial breakthrough for the specified use case. In contrast active radiometric measurements in the gamma ray spectrum have reached market maturity and are patented by several companies [15,16]. Generally, the technique requires no extensive data processing, nor it is complicated to adapt it to different geometries. The used radiation propagates as straight rays through materials with neglectable scattering and map the density along the propagation paths. The measurement itself is not directly influenced by the process temperature nor by potential hazardous substances and is therefore considered to be robust. Furthermore, a potential insulation can remain on the object because of its low mass density and the resulting neglectable influence in the data. However, the major disadvantage of the system is the radioactive source itself, which requires proper radiation protection precautions, such as personal evacuating the surrounding area during the measurements.

In the work at hand, another possibility for continuous level measurements by Guided Elastic Waves (GEW) is investigated. GEW are often also referred as Guided Ultrasonic Waves (GUW) and becoming increasingly popular for certain tasks of NDE. Their propagation in plate-like structures ensues lower geometrical damping rates than conventional bulk waves, which generally results in longer propagation distances and better signal–noise ratios [18]. Several researchers already described the possibility of estimating the fluid filling level with GEW [19,20,21,22]. In this work the GEW amplitude decay, measured at an outer vessel shell and processed by attenuation tomography is used to determine the filling distribution. This approach is promising in terms of scalability and the capability to resolve all phase transitions. Previously patented systems are admittedly also using GEW, but process only the bulk wave time of flight [23] or the change in resonance the frequencies [24], which could have significant disadvantages for bigger vessel geometries, for changing process parameters or for the accuracy.

## 2. Filling Distribution by Attenuation Tomography

### 2.1. State of the Art

Attenuation tomography originates from geophysics [25,26] to reconstruct the distribution of the anelastic attenuation factor (Q factor) in the subsurface, indicating fluid saturation or internal friction losses. Therefore, measured attenuation rates are linked to beforehand estimated wave travel paths along different discretized model cells and followed by a search for the attenuation distribution, which explains the measured data most likely. Besides the often-affiliating mathematical challenges of nonlinearity and non-uniqueness of the solution, a reliable measurement of the attenuation rates itself is delicate. Three different methods established as standard practice: Amplitude Attenuation (AA), Centroid Frequency Shift (CFS) method and Spectral Ratio (SR) [27,28]. The oldest approach of Amplitude Attenuation (e.g., in [25]) is considered to be unreliable, since for a theoretically perfect result a measured amplitude must be related to a baseline measurement without any anelastic effects. In geophysical settings, the latter is generally not obtainable directly from the measurements and therefore must be estimated by simulations or additional data processing. The other two methods (CFS and SR) require broad band sources to map the naturally occurring contrast between the attenuation rates for higher and lower frequencies.

Attenuation tomography has already been adopted for a specific NDE scenario with GEW/GUW, namely the reconstruction of localized wall-thinning on pipe-like structures [29,30]. Hereby, AA was used as input data, generated without a directly measured baseline. Instead, the undamped amplitude was assumed constant for each measurement and only amplitude peaks of the first arrivals referring to the dominant fundamental longitudinal mode were picked. This approach relies on reproducible coupling, a homogeneous structure and the absence of interference with multiple arrivals or other wave modes in the picking time frame. However, small-scale damages represent low attenuation contrasts requiring a high signal–noise contrast.

A surrounding media on the other hand is significantly affecting the propagation of GEW by leaking energy into the media as a type of bulk wave or even creating interface waves such as Stonely or Scholte waves [18]. According to linear elastic theory, the out-of-plane component of GEW is leaking energy into any media creating acoustic bulk waves. The in-plane components are less affected since the shear modulus of fluids is vanishing and therefore suppresses leaking shear waves. This behavior was theoretically described by several authors [31,32,33] and experimentally verified [34,35,36]. For isotropic metal plates it was shown that symmetric longitudinal fundamental modes S0/L(0,2) are less influenced by a potential fluid filling than antisymmetric fundamental modes A0/L(0,1), since their in-plane to out-of-plane ratio is higher [33,36]. The phase velocities of both modes are considerably stable in the medium range of frequency-thickness products. The attenuation rates and phase velocities of both modes are converging with increasing frequency-thickness product. For considerably low frequency-thickness products (e.g., steel-water within 360kHz·mm [36]) the antisymmetric mode is generating a quasi-Scholte-wave when fluids are present. The resulting high attenuation rates and the influence on the wave speed were already used for rheological measurements in the food industry [37]. Quasi-Scholte waves contain an amount of shear leakage, which can only be explained by including viscoelasticity. This phenomenon explains also the in practice often observed attenuation of shear horizontal and other wave modes, which contain a high amount of in-plane motion [32,38].

### 2.2. Principle and Discretization

The pressure vessel and the filling is assumed as a linear viscoelastic system, with a vessel wall of finite thickness. The excitation of elastic waves at the outer vessel wall and in an appropriated frequency band will result in various elastic guided wave modes, since the shear and pressure wave velocities of the vessel wall are significantly higher than the one of a potential filling. Nevertheless, the energy of such a mode is apparently damped in the presence of an additional media since its energy is converted into other wave types. The propagation of an elastic guided wave mode for a single measurement is further approximated by a ray propagating from the sender along the travel path *l* to the receiver. During this propagation, the energy of the wave mode will be degraded by the amount ΔE with the attenuation rate Q(l): (1)$$\begin{array}{cc} \Delta E & =\int_{l}Q(l)\;dl\;. \end{array}$$

In general, the attenuation consists of geometrical spreading, internal scattering and the conversion into other wave modes or wave types. For the estimation of the filling distribution particularly the latter one is of interest. As linearity is imposed, the damping from the conversion can be isolated in *Q* for arbitrary structures by additionally defining:(2)$$\begin{array}{cc} \Delta E & =10\log{}_{10}\left(\frac{E^{B}}{E^{C}}\right)\;. \end{array}$$

The measured energy degradation ΔE is therefore the difference between the energy of a baseline state $E^{B}$ without filling and a current state $E^{C}$ with a potential filling in a decadic logarithmic scale. This choice is justified to minimize errors in the numerical solutions later, since the attenuation rates are expected to be scale comprehensive. Using the ratio of energy values directly instead of subtracting raw signals beforehand, it is expected to be more stable against phase jitters or disturbances in the velocity by small-scale variations of the environmental conditions even without additional compensation. However, in contrast, this method is considered less accurate since it acts as low pass filter neglecting interference and spectral shifting effects. Both possible methods include damping effects originating from elastic and viscoelastic effects as well, if those are contained in the investigated GEW mode.

Strictly, Equation (Equation 1) must be considered non-linear because the filling distribution is influencing the GEW phase velocities and therefore the propagation path itself. As a result, linearization would require a suitable approximation such as a Taylor series, resulting in an iterative process. At each iteration, the propagation paths would have to be recalculated and therefore increasing the computational effort, the difficulty of extracting a reliable ΔE and therefore add modelling errors. To circumvent these obstacles, a horizontal separator vessel is assumed with a layered filling distribution only varying slightly along the vessel length. As a result, the filling of each cross-section is mirror-symmetric along a vertical axis parallel to the gravitational force, dividing the circumferential shell in a left and a right part with identical damping rates. This implies two advantages. First, measurements for a single cross-section would not suffer from propagation path deformation, independent of the filling distribution inside. Only the arrival times would still change, but in a more predictable manner, segregating the nonlinearity in the process of data preprocessing. Secondly, the filling in each cross-section can be described by a one-dimensional distribution Q(z) of the attenuation along the height *z* inside the vessel. Each infinitesimal height section is connected to according sections on the left and right part of the vessel shell, linking a potential propagation path to the height l=l(z).

By discretizing the height into *N* cells given by the index *j*, Equation (Equation 1) can be rewritten for *M* measurements given by the index *i* with previous assumptions as:(3)$$\begin{array}{cc} \sum_{i}\Delta E_{i} & =\sum_{i}\sum_{j}\left(Q\left(z_{j}\right)dl_{ij}\right)\;. \end{array}$$

Hereby $dl_{ij}$ is the length of a ray section inside the *j*-th cell for the *i*-th measurement. The measurements are given for various sensor-sender combinations, sufficiently covering the circumferential shell and therefore containing attenuation information along the height. The above problem can be rewritten as a linear equation system of the type:(4)$$\begin{array}{cc} d & =Fm\;. \end{array}$$

The data vector d contains the measured $\Delta E_{i}$, the operator matrix F the entries of the ray sections $dl_{ij}$ and the model vector the unknown attenuation rates $Q(z_{j})$ along the height inside the vessel. As most rays hit only a few cells along their path, the operator matrix is of sparse nature. Depending on the chosen discretization and the measurement combinations, the equation system is generally overdetermined with M≥N and the data vector is disturbed by noise. As a result, the matrix is not invertible and the model vector has to be estimated by solving an optimization problem in a chosen $L_{p}$-metric:(5)$$\begin{array}{c} \min{∥d-Fm∥}_{p}\;. \end{array}$$

To ensure uniqueness, additional requirements on the solution must be imposed. Also, constraints can increase the accuracy further. For example, since no amplifying effects are expected, it can be assumed that all entries in the solution m and the measured energy degradation in all entries of d should be non-negative. The latter can be violated in real measurements by the presence of noise.

Another possibility to increase the resolution is to extend the data space by acknowledging additional arrivals which preferably have different travel paths and therefore add information from different height sections. By using the first and second arrival for a single sender-sensor combination all height levels of a single-phase filling could be measured, as depicted by Figure 1. The linear equation systems from the *I*-st till *K*-th arrival are concluded in a weighted optimization problem of the form:(6)$$\begin{array}{c} \min{∥\begin{array}{c} \left(\begin{array}{cccc} diag(w_{I}) & \; & & \\ \; & diag(w_{\mathrm{II}}) & \; & \\ \; & \; & \ddots & \\ \; & \; & & diag(w_{K}) \end{array}\right)\left(\left[\begin{array}{c} d_{I}\\ d_{\mathrm{II}}\\ \vdots \\ d_{K} \end{array}\right]-\left[\begin{array}{c} F_{I}\\ F_{\mathrm{II}}\\ \vdots \\ F_{K} \end{array}\right]m\right) \end{array}∥}_{p}\;. \end{array}$$

Hereby each arrival has an own data vector and operator matrix, which could have a different number of row entries *M*. The optional weighting vectors $w_{I}$ to $w_{K}$ control the contribution of each arrival. Consequently, their single entries control the contribution of each measured signal to the solution.

## 3. Experimental Validation

### 3.1. Experimental Set-up

The multi-phase filling level estimation by guided waves was validated by a mock-up scale experiment, which is shown in Figure 2. The mock-up is an approximately 2.5 m long steel tank with a wall thickness of 16 mm and an inner diameter of 1 m, approximating a horizontal separator vessel without internals. One end of the mock-up is closed by a torispherical vessel head including an additional 3” ball valve with a drain. The other side contains a flange with a waterproof sealable hatch. Near this hatch a cylindrical-shaped nozzle on the top side is located, with an inner diameter of 30 cm and a height of 10 cm, measured from the top peak of the cylindrical surface as reference. It is also sealable by a waterproof hatch. The inner surface is coated by anticorrosive paint.

The outer surface of the mock-up was instrumented with two rings, each consisting of 48 circular shaped piezoelectric patch transducers of the type P876-K025 from the company PI-ceramics. Those patch transducers have a diameter of 10 mm, a thickness of 0.2 mm and an electric capacity of 3.8 nF. Each patch was glued with a two-component adhesive to the beforehand polished vessel surface. During the hardening, magnets were used to ensure a constant contact force. The transducers of each ring have an equidistant spacing of 5 cm, in relation to the outer circumference. Each ring is covering the bottom 85 cm of the inner height, whereby the upper 15 cm were uncovered due to the presence of the top nozzle at one ring. In the following, the ring near the top nozzle will be referred as ring 1 and the other one as ring 2 for the sake of clarity.

A measurement contains the transmitted signals from each sending to each receiving transducer (full matrix capture) for a certain excitation signal and a possible filling distribution. The excitation signals were Hann-windowed five cycle cosine functions (RC5) of either 50, 70, 100 or 150 kHz. To enhance the signal-to-noise ratio, each measurement was repeated 64 times and mean values were calculated afterwards. For each sample, a standard deviation was calculated beforehand and only signals with all samples within a 3.5 times standard deviation threshold were accepted. This processing step was added to remove undesired electrical noise.

Various combinations of sand-water-oil fillings were measured, which can be obtained from Table 2. Before starting the filling process an empty baseline measurement was obtained for all excited centroid frequencies. The tank was filled consecutively with an increasing water height by 10 cm per measurement. After reaching 60 cm water height an additional oil layer was introduced consecutively by 5 cm thick layers, till the total top-level reached a height of 75 cm. The oil consisted of commercially available sunflower oil representing a crude oil layer. Subsequently, the vessel was emptied and the whole process repeated with various distributions of finely grained quartzite sand, whereby the last sand filling built in with an artificial $15^{\circ }$ slope, which was fixed by an additional construction fleece. The grain size of the sand was within the range of 0.1 up to 0.3 mm. All height values were controlled by a meter stick and an ultrasonic pulse-echo device placed at the top hatch.

### 3.2. Data Processing

For the attenuation tomography, the vessel was represented by two one-dimensional height versus attenuation rate distributions. Each distribution was associated with one sensor ring. Therefore the inner height was sampled, so that the outer circumference is nearly equidistantly sampled by 1 cm cells. The deviation from a perfect equidistance results from the symmetry condition splitting into left and right circumference. The whole system implies only circumferentially acquired measurements for each ring separately, as discussed in Section 2.2. For the data preprocessing an analytic model of a steel hollow cylinder with a wall thickness of 16 mm was used. Based on the model, the time of flight, the level of waveform spreading and the attenuation due to geometric spreading were estimated beforehand, as shown by Figure 3. According to those parameters, a window function for each mode and arrival of interest was defined as:(7)$$\begin{array}{cc} \alpha (t) & =\left\{\begin{array}{cc} B(t,t_{a}) & \left(t_{a}-T_{c}\right)\geq t\geq t_{a},\\ 1 & t_{a}>t>t_{b},\\ B(t,t_{b}) & t_{b}\geq t\geq \left(t_{b}+T_{c}\right),\\ 0 & else, \end{array}\right. \end{array}$$
with the Blackmann function flanks:(8)$$\begin{array}{c} B\left(t,t_{0}\right)=\frac{21}{50}+\frac{1}{2}\cos\left(\frac{\pi (t-t_{0})}{T_{c}}\right)+\frac{2}{25}\cos\left(\frac{2\pi (t-t_{0})}{T_{c}}\right)\;. \end{array}$$

The beginning $t_{a}$ and endpoint $t_{b}$ mark the points in time, at which the estimated analytic waveform has decayed below 3% of the peak value after propagation along its path. A signal passing through in between the interval. The signal is flattened by a Blackmann function $B(t,t_{0})$ outside the interval and within the period of $T_{c}$. All other time sections are suppressed. $T_{c}$ is the characteristic period of the originally excited centroid frequency. As a result, the signal energy *E* of a certain mode arrival of a measured signal u(t) can be extracted by:(9)$$\begin{array}{cc} E & =\int_{0}^{\infty }{\left|\alpha \left(t\right)u\left(t\right)\right|}^{2}\;dt\;. \end{array}$$

This approach has the advantage of being simply adaptable to more complex numerical models and acknowledging the attenuation in a broader bandwidth than only picked peak values, to make the method more robust against waveform deformation.

The first and second arrival of the fundamental antisymmetric lamb mode were extracted. Values deriving from time sections with interference of both arrivals, meaning overlapping α(t) sections, were related to appropriated weights *w* in the equation system. For example, for a sender-sensor combination half the circumference apart from each other α(t) overlaps exactly, therefore the retrieved energy is associated with the second and first arrival with $w_{I}=w_{II}=0.5$. Since the analytic model did not contain the influence of the nozzle at ring 1, any back wall or welded seams, higher modelling induced errors are expected in the picked energy values and final solution of the optimization problem of ring 1 compared to ring 2. Due to sensor arrangement, it is expected that solution and data retrieved from the second arrival are stronger influenced by this effect.

Solutions for the solely first arrival data and combined data from the first and the second arrival were computed. In the combined solution all entries of the second arrival were penalized by additional 30% in $w_{\mathrm{II}}$. This step can be justified due to longer propagation paths and the corresponding lower energy values, which finally result in a lower signal-to-noise ratio. Also, the likelihood of interference with other modes, reflections and converted bulk waves is increasing on later arrivals. The solution was calculated with an iterative Least SquaRes solver (LSQR) of [39], which is part of the Linear Algebra PaCKage LAPACK. As a starting model an empty tank was assumed.

For each filling combination, excitation frequency and sensor ring, the solution was calculated for artificially reduced sensor networks, giving equidistant spacings of 5, 10, 20 and 30 cm. All possible combinations of shifting the network were considered likewise for each reduction step. From each single attenuation rate four different height levels were extracted, namely water-, oil-, sand- and top-level. The latter refers to the upper liquid level without distinguishing between water and oil. Based on those results, statistically validated accuracies were calculated and are given in Appendix B. The underlying attenuation thresholds for the level extraction were accordingly chosen to the first results. Those are explained in detail in Section 4.2. Therefore, oil was in the range of 7.5 dB/m till 11.75 dB/m, whereby water was above. Sand was indicated by a significant drop in the attenuation rate at least below 11.75 dB/m, but necessitated an estimated water phase above. The rules applied for all excited centroid frequencies and both sensor rings likewise.

## 4. Results

### 4.1. Data Quality

In Figure 4, exemplary signals for a single sender-sensor combination acquired at sensor ring number two, for various water-oil levels without and with a 31 cm sand phase are depicted. The fundamental symmetric longitudinal wave mode L(0,2) is arriving first due to the highest wave speed of all observed wave types. As predicted by the theory in Section 2.1 the attenuation of the L(0,2) mode is lower than for the fundamental antisymmetric mode L(0,1). L(0,1) was chosen for the further investigation due to the higher amplitude and the higher dynamic range.

With increasing filling level, the amplitude of the L(0,1) first arrival (FA) is significantly decreased. The change for the first 10 cm water and 0 cm sand filling is most significant. It implied the highest increase in travel path which is contact with water for the given sender-sensor combination. The underlying effect of a conversion into a bulk wave is also recognizable by an appearing arrival after the L(0,1) second arrival (SA). With increasing water level, the amount of converted energy is increasing and therefore the amplitude of the bulk wave. After a filling height of 40 cm and above the amount of converted energy has reached a steady state. Since, both transducers are located below this height and therefore the amount of travel path in contact with a filling does not increase further. Neglectable deviations from this observation possibly originate from the rising load pressure.

Above filling heights of 40 cm the attenuation of the SA-L(0,1) is increasing, whereas below 40 cm the effect is neglectable. For this particular sender-sensor combination, the information regarding the oil filling is resolved only in the SA. The close temporal distance between the SA and the bulk wave reveals the problem of the higher probability of interference in the time frame of the SA. For instance, for the 40 cm water filling, the additional bulk arrival generates a higher energy value than for an unfilled state, which falsifies the measured attenuation value and even switch the algebraic sign.

The presence of a sand phase is depicted in the lower row of Figure 4. Therefore, the measured filling states start with a 40 cm water level. The empty measurement in the absence of sand was added for visual convenience. The principal effects regarding the attenuation remain, even with the slightly increased attenuation rates for both fundamental GEW modes compared to measurements without sand. The fundamental difference is the apparent absence of the bulk wave in the displayed time frame. This possibly subordinated by two factors. First a slower wave speed of 500 m/s in sand than 1230 m/s in water and therefore a shifting of the arrival beyond the displayed window and secondly the high attenuation rate of the saturated non-compacted fine sand due to internal friction along the bulk wave travel path in the observed frequency range.

Regarding the phase velocities all exemplary GEW arrivals show no significant change with increasing filling level in Figure 4 even in the presence of a sand phase. A quantitative analysis of this effect for the FA-L(0,1) can be found in Figure 5. It displays the change of the FA-L(0,1) peak position for all paths compared to the baseline and finally translated into a change of the apparent group velocities Δc, for the known travel distances. Figure 5 contains all experimental acquired data which was preselected in such a way that only propagation paths along a single phase or phase transition are given. The water-sand phase transition paths passing the transition boundary were only selected once. The results substantiate that the influence of the filling on the velocity is negligible in the chosen frequency range. For the validated excitation frequencies, media and phase transition causes a change in velocity below 5 m/s for the median, below 20 m/s for the inter quartile distance and below 45 m/s for 2.7 times standard deviation. The air or empty phase in Figure 5a indicates a filling independent velocity drift below 10 m/s for all excitation frequencies. This noise possibly originates from varying environmental conditions as well as vibrational and electrical sources. The drift is within the defined window functions α(t) for the current approach of the arrival extraction and should have no significant influence on the energy estimation. For example, a possible outlier causes a drift of 75 m/s, for an original group velocity of 3222 m/s and for 100 kHz this would result in a shift of 11.6μs for the worst case along the longest possible travel path. In Comparison, the length of the characteristic period is 10μs.

Figure 6 displays the from the measurement data retrieved attenuation rates *Q*, for sensor ring two, comparable acquired and selected as Δc in Figure 5. Those attenuation rates were estimated by the measured energy decay ΔE, for all known filling distributions and propagation paths, without solving an optimization problem. Finally, all paths were selected which solely were in contact with the specific filling or phase transition. The results for the empty shell sections in Figure 6a indicate a noise level up to the 5 dB range for all investigated excitation frequencies. Implausible damping values implying an increased energy are most likely noise or interference related. The results for the water phase are given in Figure 6b. As expected according to the theory, the median attenuation rate is decreasing gradually with increasing excitation frequency from 17.4 dB for 50 kHz till 14.5 dB for 150 kHz. For the water saturated sand phase in Figure 6c the results are similar, with a significantly comparable quantitative congruence. This would imply difficulties in mapping the contrast between the two media at the first level of deduction. On the other hand, the data for a single-phase transition between water and sand has a significantly lower median attenuation rate (Figure 6d) and deviation as for both solely phases. This indicates that another type of interaction is induced due to phase transition, which is otherwise not influencing Δc significantly, as can be seen by comparing Figure 5d with Figure 5b,c.

The underlying data of Figure 5d are resolved as blockwise defined propagation distances of all arrivals along a water-sand boundary in Figure 7. This graphic substantiates the assumption for each excitation frequency separately that the phase transition itself is mainly influencing the attenuation rate. With increasing propagation distance, the attenuation rate is converging to the level of the purely sand or water filling itself, whereas for close distances the effect of the transition is dominating and lowering the attenuation rate. Based only on the data driven evaluation, the reason for phase transition effect is up to speculation. The effect is probably based on a kind of surface waves as Scholte waves or a reflection, which are partially reintroducing the energy at the phase transition, resulting in locally reduced attenuation rates. All above explained effects can be observed for sensor ring number one as well in Figure A1, Figure A2 and Figure A3. Those are given in Appendix A. A similar reliable evaluation of the oil phase is not available, due to the limited amount of data points. For each excitation frequency a lower decade of points is not comparable to multiple thousand points as for sand-water or the sand phase, in detail.

### 4.2. Attenuation Mapping

In the following section, various results of the attenuation tomography/tomometry are displayed and discussed regarding their characteristic behavior and their quality. Quantitative results can be found in Appendix B and give a statistically validated accuracy of the technique for the given geometry. Those are also cited during the following discussion when accuracies of the level estimations are mentioned.

Exemplary results for 100 kHz excitation frequency, 15 cm sensor spacing and three water filling levels (10, 40 and 60 cm) are given by Figure 8. The calculated attenuation rates of the filling along the height axis are displayed in the upper row, three-dimensional representations of above results can be found in the lower row. Latter were estimated from the given attenuation rates at the sensor rings and by the linear interpolation along the length axis. The color scheme with defined limits was arbitrarily chosen after the first results and used for all calculations likewise to ensure comparability. Oil has attenuation rates in the range of 7.5 dB/m till 11.75 dB/m and is represented by a beige brown coloring in the three-dimensional graphic. Whereby, water has attenuation rate above 11.75 dB/m and is represented by the blue coloring. Sand is indicated by a significant drop in the attenuation rate at least below 11.75 dB/m as seen in the results below. Therefore, the sand phase has at least beige brown till gray coloring. Note that the three-dimensional representations visualize only the interpolated results of the FA-solutions in the following figures. The gray dots indicate the unused and the black dots the used sensors for calculating the attenuation rates artificially emulating a desired sensor spacing.

Due to the possibly low attenuation contrast between water and oil false indications could occur. Parts of the water layer can be estimated as oil, as shown for the measurements without any oil phase in Figure 8a–c. Otherwise parts of an oil layer could be identified as water, as indicated in Figure 11b,c. In general, the error has a systematic drift, underestimating the water level and overestimating the oil level. This behavior validated by the algebraic signs in the calculated error rates in Table A3 and Table A4. Nevertheless, the median error of the water level is 0 cm for ring two and 4 cm for ring one for a sensor spacing of 15 cm and an excitation frequency of 100 kHz. The error distribution is in the range of −5 to 8 cm and −4 to 9 cm given by the boundaries of the 32nd and 68th-percentile, respectively. The accuracy remains in the same range for the investigated sensor spacings.

The oil level in Figure 9 is estimated with high accuracy for the 10 cm (Figure 9b) and 15 cm (Figure 9c) filling at both rings. The 5 cm case in Figure 9a displays an overestimated oil layer, at least for sensor ring number two. Generally, the median error of the oil layer for ring number two (Table A4) is in the range of down to −4 cm, coinciding with the above observation of tendentially underestimating the oil thickness for the FA input data. By combining FA and SA data the median error is lowering till ±1 cm for nearly all cases investigated. In contrast, ring one has a median error in the range between −3 up to 1 cm for the FA data, whereby the median error is decreasing for additional SA data. This is possibly caused by the structural influence of the nozzle, which was not acknowledged by the model and therefore in the calculated propagation paths. This is resulting in picking errors in the energy extraction processing and secondly assigning measured attenuation values incorrectly to various height sections in mapping algorithm. This effect manifests especially in the SA data, since the sensor rings are not fully instrumented around the whole circumference in the upper height sections. Therefore, most SAs are along the upper height sections, were the oil level was located for the investigated filling states. In case of ring two, this adds coherent information, whereas for ring one the presence of nozzle adds incoherency to the SA data. This circumstance manifests itself in nearly every error distribution, for all investigated combinations, of all phases of ring one in Appendix B, even if the median accuracy remains considerably stable. For example, this accuracy difference is also observable for the estimated top levels, as detailed given by Table A1. The top-level is the upper fluid level without distinguishing between the fluid content. Therefore, the accuracy is significantly higher than for oil or water separately. The median accuracy is −1 cm and the inter 68th-percentile range is less than 9 cm for ring two even for a sensor spacing of 30 cm. For ring one the results are comparable, at least for the FA solution.

Oscillations in the solution in the lower height range are observable for all cases of filling combinations: water (Figure 8), oil-water (Figure 9) and sand (Figure 10). Those are visible as spike-like increased or decreased attenuation rates, nearly at the bottom of the vessel. Those are most likely consequences of the numerical conditioning of the discretization, errors in the input data and the incomplete instrumentation along the circumference. Therefore, most of the FA travel paths are along the lower height sections, adding a high amount of probably incoherent information to the section.

Sand fillings, as displayed by Figure 10, cause a significant drop in the attenuation rate compared to fluids. This implies a lower conversion rate which was not validated by other models or literature beforehand. Concluding with the argumentation of data quality analysis in Section 4.1, this is probably due to an effect at the phase transition itself, rather than the attenuation along the water or sand filling. In the case of the 19 cm (Figure 10a) and the 31 cm (Figure 10b) filling, the solution reveals significant oscillations in the lower centimeter height range, exposing a rapid increase in the attenuation rate. Those could originate from the previously described inconsistencies in the FA arrival. Additionally, the paths which remain in the sand phase could add incoherency, due to their higher damping rate as was already discussed for Figure 6. However, the predefined picking rules excluded such oscillations from the height level estimation, if those remain below 11.75 dB/m. The median error of the sand level estimation itself is 0 cm for almost all validated excitation frequencies, sensor spacings and for both sensor rings as given by Table A2. The inter 68th-percentile distance is around 4 cm for frequencies up to 70 kHz and 15 cm sensor spacing. With increasing frequency and sensor spacing the inter percentile distance is growing till 10 cm and 18 cm in some cases.

The tables in Appendix B contain the quantitative difference for the tested sensor spacings. A qualitative impression for the difference in sensor spacing can be obtained from Figure 11. The estimated attenuation rates in the first row decrease in smoothness with decreasing sensor spacings. For the 5 cm sensor spacing in Figure 11a this implies a smooth change in the attenuation behavior of the GEW during phase transitions. However, the smoothness itself is not necessarily a result of the underlying physics and rather an effect of the used LSQR algorithm and the smoothness constraints to guarantee uniqueness of the solution itself. Therefore, the 15 cm sensor spacing in Figure 10c is apparently more agreeing with the expected physical behavior. However, the lower smoothness has apparent disadvantages in the interpolation process. As a consequence, the visual leap in the sand slope is increasing with the sensor spacing, even though the single attenuation distributions are comparable accurate.

## 5. Discussion and Conclusions

In the work at hand, the estimation of multi-phase height levels by attenuation tomography with guided elastic waves was realized. The investigated case was a small-scale horizontal separator vessel filled with various distributions of sand, water and oil, whereby piezoelectric transducers were mounted on the outer shell. The results show a reliable and accurate system regarding the top and sand level with accuracies in the centimeter range for a sensor spacing of 15 cm. The ability of exactly distinguishing between water and oil suffers from higher inaccuracies due to low contrast in the attenuation rate. The median error is in the lower centimeter range, whereby the 32nd to 68th inter percentile distance is in the decadic centimeter range. However, already existing practicable solutions for estimating fluid phase transitions could be added to enhance the results further by defining additional constraints in the optimization problem.

The proposed technique is based on attenuation amplitude measurements by picking a specified wave mode and its arrivals and set them in relation to a baseline measurement. Therefore, a model-based approach was used which can be adopted to more complex geometries including nozzles or even internals. In the investigated frequency range, the filling showed no significant influence on the wave speed. Based on the analysis of the measurement data the attenuation rate of saturated sand and water were in the same range, whereby a phase transition between both phases added an unexplained mechanism, which caused a significantly decreasing attenuation rate. The proposed algorithm was able to resolve the phase transition. However, the appearance of bulk waves was observed for fluid fillings, whereas the presence of sand apparently damped the direct bulk arrivals. This behavior represents another possibly measurable contrast or addable information to enhance the accuracy of the system further. Basic assumptions regarding the filling distribution and sensor arrangement were made for the attenuation tomography. Those resulted in symmetry conditions which enabled a reduction of the model space to an independent one-dimensional optimization problem per sensor ring. Consequently, the calculated attenuation distributions were one-dimensional cross sections which coincide with the term tomometry introduced by [7]. By combining the results from two sensor rings at different locations it was possible to resolve an artificial $15^{\circ }$ sand slope and to estimate the fluid boundaries at the same time without further interconnections between the data. This and the excluded geometrical features at one of the used sensor rings indicate the robustness and reliability of the technique.

The investigated frequency-thickness product ranged from 800 to 2400kHz·mm and showed no significant influence on the accuracy for all fillings except for sand. However, this was significantly below the expected rate of 360kHz·mm for the appearance of quasi-Scholte waves at a steel-water boundary [36]. However, it seems possible to use the effect in further investigations to enhance the measurement contrast or to resolve rheological properties of oil or wax in the vessel.

For the investigated vessel geometry with an inner diameter of 1 m and a wall thickness of 16 mm, the 15 cm equidistant sensor spacing along the outer shell quantified as a reliable trade-off between accuracy and minimal instrumentation effort. For higher spacings, the accuracy is decreasing for the given tasks, whereas the pure top-level estimation was reliable even for investigated sensor spacings of up to 30 cm. In the current data processing only first arrivals and the combination of first and second arrivals of a specific guided wave mode were used. The latter gave no significant improvement of the accuracy for most investigated fillings. This circumstance was probably substantiated due to the lower signal–noise ratio of the SA arrivals. The exception was the estimation of the oil level, which possibly originates from the incomplete circumferential instrumentation at the upper 20 cm of the inner height of the test specimen. This results in a higher amount of SA travel paths along the upper height levels, where the oil levels were located for the investigated filling states. Consequently, for a full circumferential instrumentation this exception is expected to converge to an insignificant amount.

The system could be adopted to vertical separators, whereby a single vertical line long the vessel wall seems like a logical choice for a sensor arrangement. This would imply a simpler discretization scheme with a purely linear dependence on the filling height, which could possibly lead to a better numerical stability by a lower condition number. On the other hand, the additional information given by multiple arrivals would be not available.

Upcoming work should address the stability and influences in case of changes in the process parameters as temperature or pressure. The possibility of estimating residual wax layers or mixed phases as foam or emulsion should be evaluated. Additionally, the possibility of including structural features as internals or nozzles in the propagation paths by further processing such as ray tracing or eikonal solvers and its capable improvements in accuracy could be assessed. A further theoretical question is the underlying wave propagation mechanism along the water-sand phase transition for the investigated geometry and frequency range, which is causing the significant drop in the attenuation rate. The overall observed high data quality is a complied requirement for more elaborate processing schemes as full waveform inversion, which could acknowledge multiple modes at the same time, include complex mechanisms along phase transitions and eventually result in elastic and viscoelastic properties rather than in abstract attenuation rates.

## Figures and Tables

**Figure 1 sensors-21-00179-f001:**
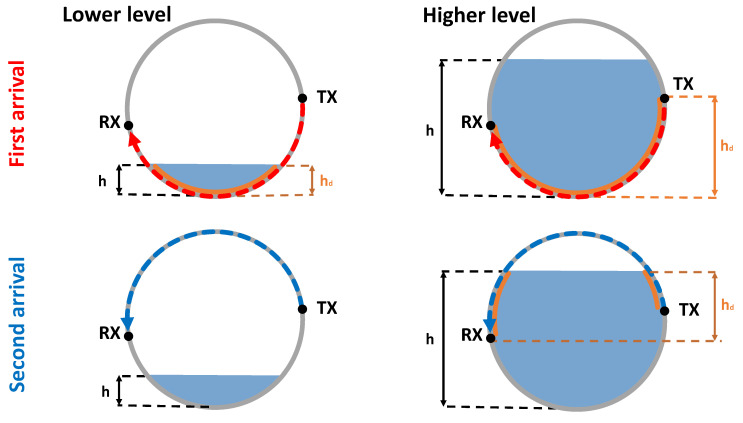
Exemplary first arrival travel path (FA, red) and second arrival travel path (SA, blue), for two different filling heights below (**left row**) and above (**right row**) the participated sender (TX) and sensor (RX). For each combination the travel path length with occurring energy conversion caused by the filling is marked orange. The resulting height section covered by the data is $h_{d}$ whereby the exact filling height is given by *h*.

**Figure 2 sensors-21-00179-f002:**
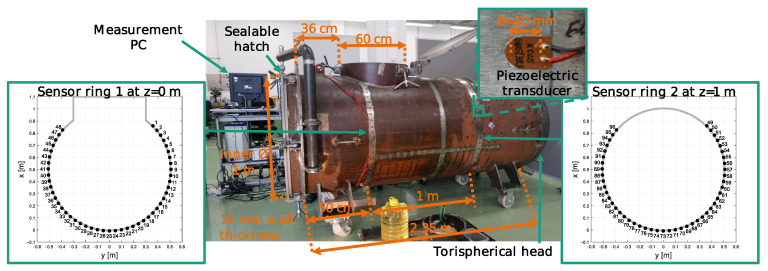
Experimental mock-up used to assemble guided elastic wave data of various filling states. The **left**- and **right**-side display schematic cross sections of the vessel with the exact sensor placement (black dots) for the two sensor rings with the global sensor numbering from 1 to 96.

**Figure 3 sensors-21-00179-f003:**
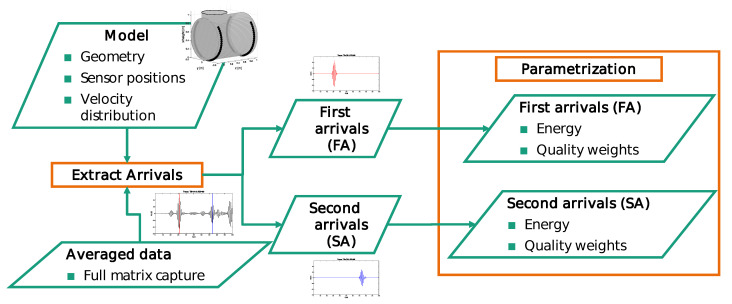
Schematic overview on the parametrization of the measured guided wave data.

**Figure 4 sensors-21-00179-f004:**
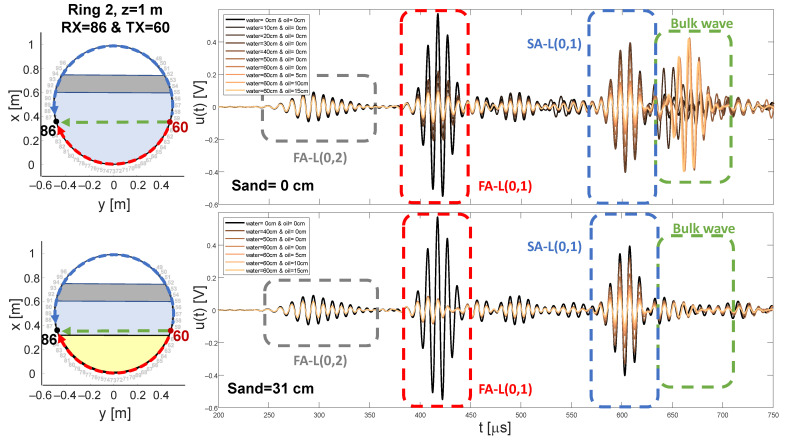
Exemplary signals for excitation with 100 kHz at transducer number 60 and received at transducer number 86, at the sensor ring number two for various filling states without (**first row**) and with a 31 cm sand filling (**second row**). The travel paths of first (red, FA) and second arrival (blue, SA) are color coded.

**Figure 5 sensors-21-00179-f005:**
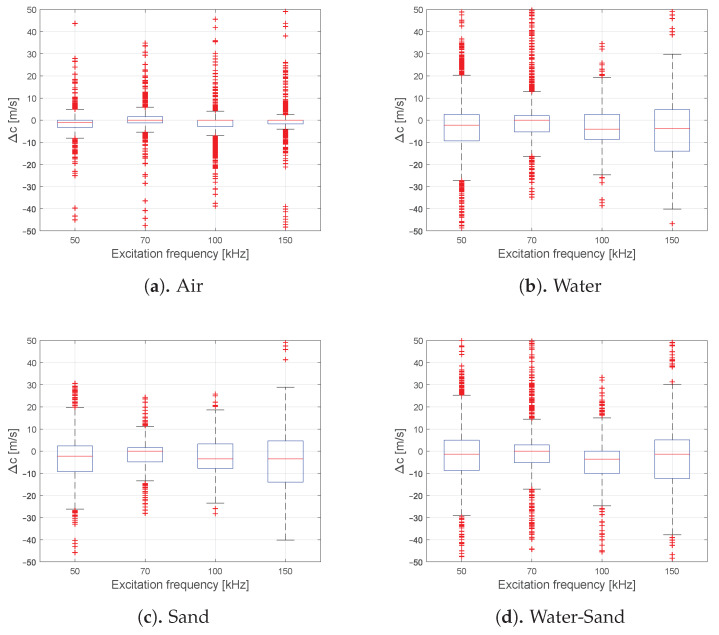
Change of the estimated group velocities Δ*c* for all acquired experimental data compared to its reference. The data contains all measured circumferential FA-L(0,1) of ring number two and specified for each excitation frequency separately. Additionally, the data are selected for propagation paths solely along the given phase (air, water or sand) or phase transition (water-sand).

**Figure 6 sensors-21-00179-f006:**
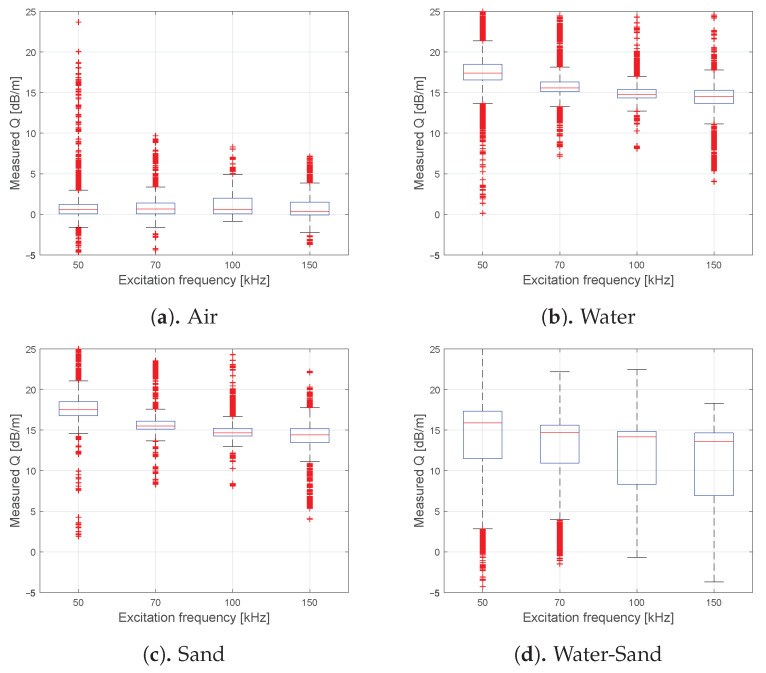
Estimated attenuation rates *Q* along each phase for the measured energy decays and the known travel distances. The data contains all measured circumferential FA-L(0,1) of ring number two and is specified for each excitation frequency separately. Additionally, the data is selected for propagation paths solely along the given phase (air, water or sand) or phase transition (water-sand).

**Figure 7 sensors-21-00179-f007:**
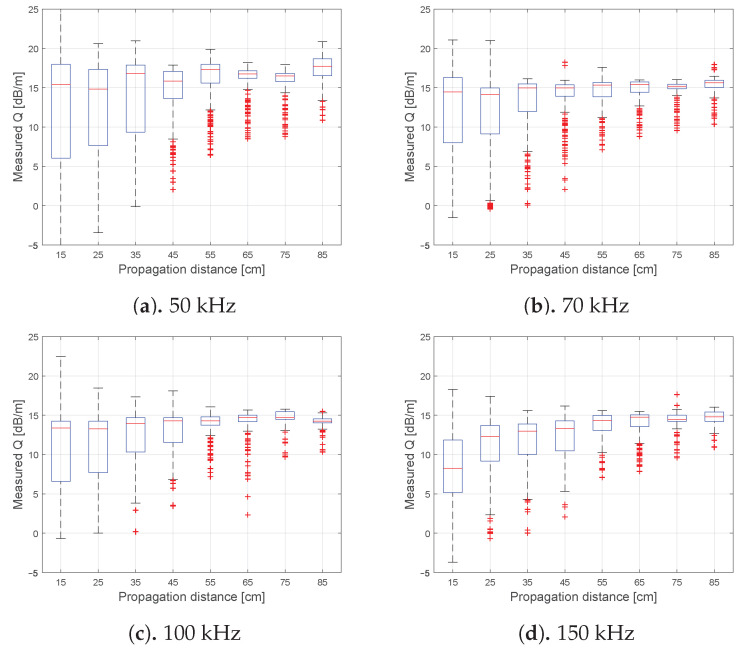
Estimated and sorted attenuation rates *Q* along the water-sand transition. The data contains all measured circumferential FA-L(0,1) of ring number two and the four excitation frequencies. The data is sorted and selected as blocks according to the propagation distance of each arrival.

**Figure 8 sensors-21-00179-f008:**
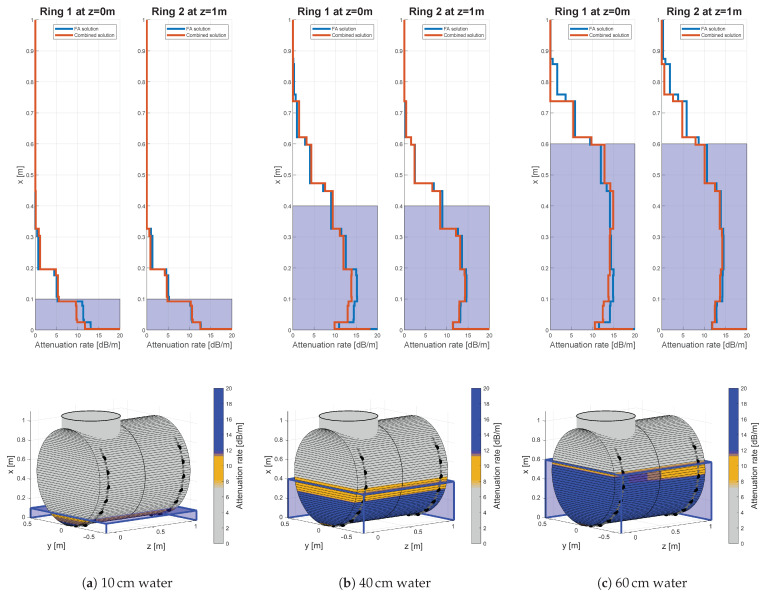
Results for various solely water fillings for 100 kHz excitation frequency and 15 cm sensor spacing.

**Figure 9 sensors-21-00179-f009:**
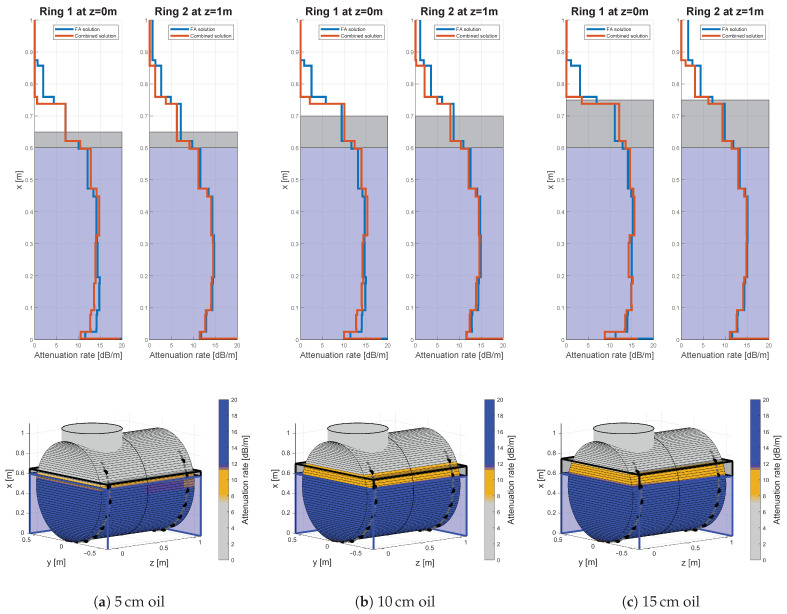
Results for three different oil fillings, 60 cm water filling, an excitation frequency of 100 kHz and 15 cm sensor spacing.

**Figure 10 sensors-21-00179-f010:**
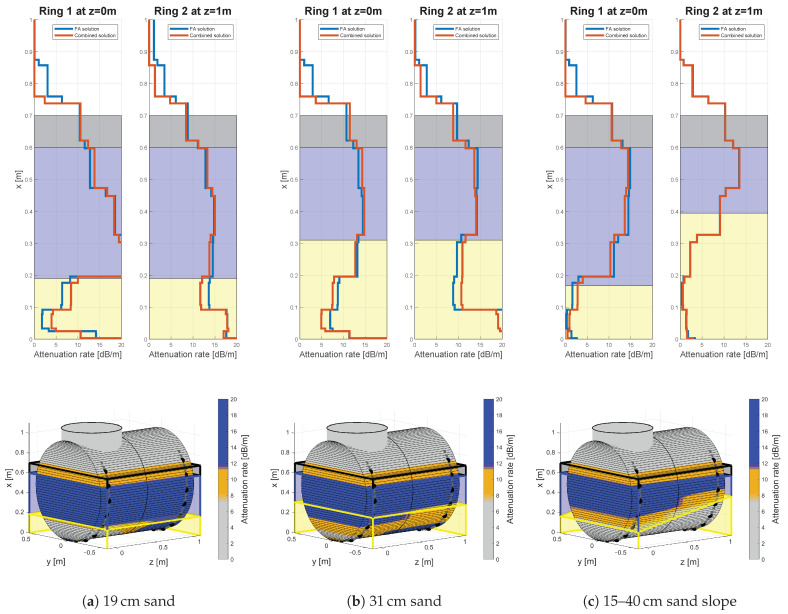
Results for three different sand fillings, 10 cm oil, 60 cm water, an excitation frequency of 100 kHz and 15 cm sensor spacing.

**Figure 11 sensors-21-00179-f011:**
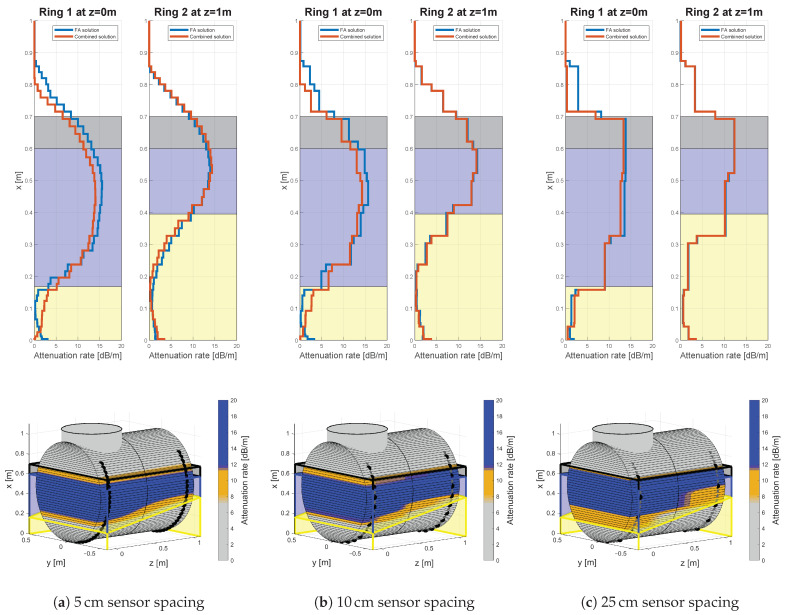
Results for three different equidistant sensor spacings, 10 cm oil, 60 cm water, a 15° sand slope and an excitation frequency of 100 kHz.

**Table 1 sensors-21-00179-t001:** Basic available continuous level measurement techniques with the ability to resolve a phase transition ✔, partly resolve it under certain circumstances (✔) or with the inability to resolve it ✘. Additional conditions as wax, emulsion or foam can disturb ✘ or partly disturb (✘) the estimation of the specific phase transition. In case of guided elastic waves the ? marks the unavailable experience with additional conditions together with the partial design of the experiment in this work.

Principle	Instrumentation	Fluid-Gas	Fluid-Fluid	Fluid-Solid	Disturbances		
					**Wax**	**Emulsion**	**Foam**
Pressure sensor	Partly inside	✔	✔	✘		(✘)	(✘)
Displacer	Partly inside	✔	✔	✘		✘	✘
Ultrasonic pulse-echo	Inside	✔	(✔)	(✔)		✘	✘
Pulse Radar	Inside	✔	✔			(✘)	
Guided radar	Inside	✔	✔	(✔)	(✘)		
Multi-electrode capacity	Inside	✔	(✔)				
Electric capacity tomography	Outside	✔	✔	(✔)			
Gamma ray	Outside/Inside	✔	✔	✔			
Guided elastic waves	Outside	✔	(✔)	✔	?	?	?

**Table 2 sensors-21-00179-t002:** Measured filling state combinations marked by ✔. Note that the heights of the oil phases are related to the 60 cm water filling, while the water height is giving the top-level even in the presence of a sand phase.

Height [cm]		Sand			
		0	19	31	Slope: 15–40
**Water**	0	✔			
	10	✔			
	20	✔			
	30	✔	✔		
	40	✔	✔	✔	✔
	50	✔	✔	✔	✔
	60	✔	✔	✔	✔
**Oil**	5	✔	✔	✔	✔
	10	✔	✔	✔	✔
	15	✔	✔	✔	✔

## Data Availability

The data presented in this study are available on request from the corresponding author. The data are not publicly available due to a legal contract between the *Fraunhofer Society for the Advancement of Applied Research* and *Equinor ASA*.

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
