# Peer review of "Estimation of the Filling Distribution and Height Levels Inside an Insulated Pressure Vessel by Guided Elastic Wave Attenuation Tomography"

_sensors, 2020, doi:10.3390/s21010179_

Round 1

Reviewer 1 Report

The paper deals with a very interesting problem and the design of experiments, the methodology used and the results obtained are of good quality.

But the writing and presentation of the results and the discussions must be significantly improved before the paper can be accepted for publication.

The major issues are: The Figure 5 and the explanation given is not very clear. A different kind of plot may be better suited for this. But if the whisker plot is persisted the explanation has to be improved radically with point by point reference to the observations and the reasons provided. Listing the exact location of focus on the plot by highlighting the case or marking with different color circles may be useful.

Similarly the discussions for the Figures 6, 7, 8, and 9 need to be improved as well. Firstly the discussion have to be structured. Then the similarities and differences in the plots compared (3 cases in the same figure) have to be explained in detail. Also they can be discussed in relation to previous cases.For instance, Figure 8 attenuation rate for a and b are considerably different than for the figure c. The discussion has to be carried out. Many such interesting observations can be made in each of the figure with regards to the attenuation at the fluid interfaces which is seen in some cases but not in all.
Apart from this several times, a reason for a peculiar observation has been made without citing a reference or giving a complete discussion. For example, the presence of nozzle leading to more uncertainty needs to be explained better. There are other such instances where explanations and discussions are needed.

Another major issue, is the applicability of the solution to real application. As the ratio based technique is used, rather than the baseline subtraction, how will the effects of reflections from other structural features on the structure be accounted for? A discussion on the application to real structures in the conclusions will be useful as well

In addition to these major changes, adding a schematic of the sensor placement in addition to the photo of the setup may improve the understanding.

Also, the literature review needs to be expanded to include discussions of more recent articles. Most of the cited articles are more than 5 years old.

Reviewer 2 Report

The article "Estimation of the Filling Distribution and Height Levels Inside an Insulated Pressure Vessel by Guided Elastic Wave Attenuation Tomography" describes a novel technique to estimate high levels inside insulated pressure vessels. The research is well done, its presentation is very clear and it reads fluently.

I only have few minor comments:

Page 4, Section 2.2; the main assumption is the filling is a linear viscoelastic system; however water does not fulfill this condition. Can you address this issue in the article? How the measurement is affected by this?

Figures 6-9  typo on the labels "Combinded solution"

Page 15, line 408 "By combing"

Reviewer 3 Report

This article is of good quality and it is hard for me to find any place for improvement. It may be printed in a delivered form. 

Round 2

Reviewer 1 Report

The changes made in the paper are satisfactory.

Congratulations on an excellent paper.